# Visualizing the Spatial Distribution of *Arctium lappa* L. Root Components by MALDI-TOF Mass Spectrometry Imaging

**DOI:** 10.3390/foods11243957

**Published:** 2022-12-07

**Authors:** Lingyu Li, Zhichang Qiu, Mingdi Jiang, Bin Zhang, Qiang Chen, Chaojie Zhang, Zhenjia Zheng, Xuguang Qiao

**Affiliations:** 1Key Laboratory of Food Processing Technology and Quality Control of Shandong Higher Education Institutes, College of Food Science and Engineering, Shandong Agricultural University, 61 Daizong Street, Tai’an 271018, China; 2Department of Chemistry, University of Massachusetts Amherst, 710 North Pleasant Street, Boston, MA 01003, USA; 3Shandong Burdock Medical Research Institute Co., Ltd., 3230 Chongde 8th Avenue, Dezhou 253084, China

**Keywords:** *Arctium lappa* L., MALDI-TOF mass spectrometry imaging, chemical components, spatial distribution

## Abstract

This study is aimed at developing novel analytical methods to accurately visualize the spatial distribution of various endogenous components in *Arctium lappa* L. (*A. lappa*) roots, and to precisely guide the setting of pre-treatment operations during processing technologies and understand plant metabolism process. The matrix-assisted laser desorption/ionization time-of-flight mass spectrometry (MALDI-TOF MS) imaging technology was used for visual demonstration of the in situ spatial distribution in *A. lappa* roots. This work consisted of four steps: matrix selection, section preparation, matrix coating, and MALDI-TOF MS imaging analysis. Consequently, eight saccharides, four caffeoylquinic acids, four flavonoids, six amino acids, one choline, and one phospholipid were imaged and four unidentified components were found. Saccharides were distributed in the center, whereas caffeoylquinic acids and flavonoids were mainly present in the epidermis and cortex. Furthermore, amino acids were mainly detected in the phloem, and choline in the cambium, while phosphatidylserine was found in the secondary phloem and cambium. This study demonstrated that MALDI-TOF MS imaging technology could provide a technical support to understand the spatial distribution of components in *A. lappa* roots, which would promote the processing technologies for *A. lappa* roots and help us to understand the plant metabolism process.

## 1. Introduction

*Arctium lappa* L. (*A. lappa*), commonly named as burdock, belongs to the family Asteraceae. It has been widely used as a traditional medicinal herb and edible root vegetable for centuries [1]. The seeds, leaves, flowers, and roots of *A. lappa* are consumed, and its root is the main part used for nutritional and pharmacological purposes. *A. lappa* root exhibits excellent antibacterial and antifungal [2], antioxidant [3], anti-inflammatory [4], and anti-atherosclerosis [5] properties. These functional properties are attributed to various nutritional components, including polysaccharides, phenolic acids, flavonoids, vitamins, lignans, and polyacetylenes [6,7,8].

At present, there are various analytical methods that can be used to identify endogenous components in *A. lappa* roots, such as high-performance liquid chromatography-mass spectrometry (HPLC-MS) [9,10], gas chromatography-mass spectrometry [11] and nuclear magnetic resonance (NMR) [12]. However, these methods have some drawbacks, including complex pre-treatment processes and lengthy consumption, which may lead to the degradation and pollution of partial components. In particular, the spatial information regarding the locations of nutritional components in the roots of *A. lappa* was normally lost over the pre-treatment process, which would not guide its accurate extraction and precision processing.

Matrix-assisted laser desorption/ionization time-of-flight mass spectrometry (MALDI-TOF MS) imaging technology is a powerful tool used to analyze spatial locations of macromolecules without any extraction, purification, or labelling technique [13]. MALDI-TOF MS imaging has been successfully applied to endogenous components’ distribution profiling [14] and metabolic process analysis [15]. Recently, this technology has been introduced to the food science field to visualize spatial distributions of metabolites [16], explore complex defense mechanisms [17], and compare plant metabolite changes at different maturity stages [18]. In previous studies, MALDI-TOF MS has been applied to detect the components of *A. lappa*; however, so far there has been no report regarding mass imaging of *A. lappa*. Liu et al. [1] analyzed caffeoylquinic acids and lignans from roots and seeds in *A. lappa* using both HPLC-MS and MALDI-TOF MS methods, in which 80% methanol was used to extract the sample. Many lignans in extract seed samples were identified and no caffeoylquinic acids were detected in seeds and roots of *A. lappa* using MALDI-TOF MS technology, while their spatial information was destroyed. Furthermore, MALDI-TOF MS was also applied to detect the saccharides of *A. lappa* roots [5]. Three saccharides with different degrees of polymerization (DP2, DP3, and DP4, [M + Na]^+^) in the crude extracts were determined by sonication with 0–60% (*v*/*v*) ratio of ethanol, while the spatial information of saccharides in *A. lappa* were destroyed as well. Most of these studies mentioned above focused on the crude extract of *A. lappa* and no spatial distribution information of endogenous components was provided. Here, it is proposed that the components could be located by the spatially resolved spectroscopy approach, which would provide important value for exploring novel analysis methods in complex food matrices.

This study is aimed at developing novel analytical methods to accurately visualize the spatial distribution of various endogenous components in *A. lappa* roots. This study is the first report on the composition and distribution of various endogenous components in *A. lappa* roots by using MALDI MS imaging. In this work, six traditional organic matrices assisting MALDI-MS analysis were compared for nine representative compounds (L-arginine, proline, phenylalanine, caffeic acid, quercetin, chlorogenic acid, ursolic acid, 1,5-dicaffeoylquinic acid, and 1,3,5-tricaffeoylquinic acid) abundant in *A. lappa* roots, and the optimal matrices were chosen. Specifically, using optimized MALDI-TOF MS matrices, the root sections were prepared by matrix coating and analyzed using MALDI-TOF MS imaging technology. The results are expected to analyze the spatial distribution information of different endogenous components in *A. lappa*, and to guide the food processing technology of *A. lappa* roots and promote the understanding of the plant metabolism process.

## 2. Materials and Methods

### 2.1. Samples and Reagents

Fresh *A. lappa* roots of *Baiji* variety (Chinese name) and *Yanagawa-riso* variety were purchased from Linyi city (Shandong, China).

α-Cyano-4-hydroxycinnamic acid (CHCA, pure 99.0%), sinapic acid (SA, pure 99.0%), 2′,5′-dihydroxyacetophenone (DHAP, pure 97%), 9-aminoacridine (9-AA, pure 99.5%), N-(1-naphthyl)-ethylenediamine dihydrochloride (NEDC, pure 98.0%), and trifluoroacetic acid (TFA) were purchased from Sigma-Aldrich (ST. Louis, MO, USA). L-arginine, proline, phenylalanine and 2,5-dihydroxybenzoic (DHB, pure 99.0%) were obtained from Shanghai Aladdin Biochemical Technology Co., Ltd. (Shanghai, China). Caffeic acid, quercetin, chlorogenic acid, ursolic acid, 1,5-dicaffeoylquinic acid, and 1,3,5-tricaffeoylquinic acid (pure 98.0%) were provided by the Shanghai Yuanye Biological Technology Co., Ltd. (Shanghai, China). HPLC-grade acetonitrile (ACN) and methanol (MeOH) were supplied by Merck (Darmstadt, Germany). Optically transparent indium-tin-oxide (ITO) coated glass slides were obtained from Huanan Xiangcheng Technology Co., Ltd. (Shenzhen, China). The optimum cutting temperature (OCT) compound was supplied by SAKURA (Torrance, CA, USA).

### 2.2. Matrix Selection

The matrix plays important roles in separate components in samples, absorbs laser energy effectively, and provides reactive ions that ionize the samples [19]. Matrices suitable for MALDI MS imaging analysis of endogenous components should have several features, including strong optical absorption to ensure high ionization efficiency, minimum matrix background interference, and homogeneous co-crystals between the matrix and the test analytes [20]. For traditional organic matrices, the detection and imaging of low-molecular-weight substances (*m*/*z* 100~600), such as caffeoylquinic acids, flavonoids, and amino acids, are more susceptible to producing background interference from the matrix. Therefore, selecting the appropriate matrix is important for the successful detection of low-molecular-weight substances in real samples.

In order to improve the detection sensitivity, the performance of three traditional organic matrices (DHB, CHCA, SA) assisting MALDI-MS analysis were compared for three representative compounds (L-arginine, proline, phenylalanine) abundant in *A. lappa* roots in the positive ion reflector mode and the other three matrices (DHAP, 9-AA and NEDC) assisting MALDI-MS analysis were compared for six representative compounds (caffeic acid, quercetin, chlorogenic acid, ursolic acid, 1,5-dicaffeoylquinic acid, and 1,3,5-tricaffeoylquinic acid) abundant in *A. lappa* roots in the negative ion reflector mode. The optimal matrices were chosen according to the intensity of target analytes and the background interference of matrices in mass spectrometry.

Firstly, nine representative compounds were prepared, including L-arginine (2 mM, dissolved in ultrapure water), proline (2 mM, dissolved in ultrapure water), phenylalanine (2 mM, dissolved in ultrapure water), caffeic acid (6 mM, dissolved in MeOH), quercetin (6 mM, dissolved in MeOH), chlorogenic acid (6 mM, dissolved in MeOH), ursolic acid (6 mM, dissolved in MeOH), and 1,5-dicaffeoylquinic acid (6 mM, dissolved in MeOH), 1,3,5-tricaffeoylquinic acid (6 mM, dissolved in MeOH). The matrix preparation process was based on previous studies with moderate modification [21,22]. Specifically, DHB, CHCA, SA, DHAP, 9-AA and NEDC were prepared in 0.1% TFA buffer (ACN/water, 1:1, *v*/*v*) at 10 mg/mL, respectively, as MALDI matrices. Subsequently, 1 μL of sample solution was transferred onto the stainless-steel ground plate and then dried in a vacuum dryer at 25 °C. After the sample solutions were dried, 1 μL of matrix solution was deposited on the top of the sample layer and then dried again.

The dried ground steel plate covered with samples and matrices was analyzed using a Rapiflex MALDI Tissuetyper™ TOF/TOF MS (Bruker Daltonik GmbH, Bremen, Germany) equipped with a smart beam Nd:YAG 355 nm laser. The mass spectrometry analysis followed our previous method, with minor modifications [22]. A mixture of DHB and CHCA standards were used for calibration of the mass analyzer, and mixed DHB/CHCA (1:1, *w*/*w*) solution was prepared in 0.1% TFA buffer (ACN/MeOH/water, 70:25:5, *v*/*v*/*v*) at 20 mg/mL [23]. The intensity and the global attenuator offset were set to 70% and 15%, respectively, and the repetition rate and frequency were 1000 Hz and 5000 Hz, respectively. The raw spectra were collected in positive reflector mode with the *m*/*z* range of 0–1200 and in negative reflector mode with the *m*/*z* range of 100–1300. The obtained mass spectra were analyzed using FlexAnalysis Version 4.0 (Build 9) software (Bruker Daltonik GmbH, Bremen, Germany).

### 2.3. Section Preparation of A. lappa Roots and MALDI-TOF MS Imaging

After washing, the fresh *A. lappa* roots were cut into 0.3 cm pieces and stored in the refrigerator at −80 °C. Prior to sectioning preparation, the *A. lappa* roots were placed in the refrigerator at −20 °C for 20 min to equilibrate temperature and mounted onto a cryostat sample holder using OCT compound. The mounted samples were then sectioned into 16 μm sections using CryoStar NX50 NOVPD cryostat microtome (Thermo Scientific, Bremen, Germany), and the sections were thaw-mounted onto ITO-coated glass slides. Finally, the slides coated with samples were dried in a vacuum desiccator for 30 min. Subsequently, dried *A. lappa* root tissue sections were transferred in a sprayer for matrix coating using a HTX^TM^-Sprayer^TM^ (HTX Technologies, Carrboro, NC, USA). The spraying parameters were as follows: The flow rate of the sprayer was set to 0.03 mL/min at 60 °C, and the track speed was set to 1200 mm/min. The moving distance of the nozzle was 3 mm, the number of spraying cycles was 18, and the spraying density of the optimal matrix on the slice was approximately 0.0025 mg/mm^2^. After matrix spraying, the mixture of DHB/CHCA (1:1, *w*/*w*) solution was dropped on the ITO glass slides where there were no sections of *A. lappa* roots for molecular weight correction [23].

Finally, the ITO glass coated samples were dried in a vacuum desiccator for subsequent MALDI-TOF mass imaging analysis. MALDI MS imaging data were acquired in positive reflector mode with the *m*/*z* range of 80–3000 and in negative reflector mode with the *m*/*z* range of 80–2000 using Rapiflex MALDI Tissuetyper™ TOF/TOF MS imaging. The spatial resolution of the tissue sections was 200 μm, and the repetition rate, frequency, intensity, and global attenuator offset were 1000 Hz, 5000 Hz, 70%, and 15%, respectively. The obtained mass spectra were analyzed using FlexImaging 5.0 (Bruker Daltonics GmbH, Bremen, Germany) and SCiLS Lab2018b software (Bruker Daltonik GmbH, Bremen, Germany).

### 2.4. Statistical Analysis

Data were expressed as mean ± standard deviation (SD). A one-way analysis of variance was performed by ANOVA, followed by Duncan’s test and Tukey’s test using SPSS (Version 28.0, Armonk, NY, USA), where *p* < 0.05 was assumed to be statistically significant.

## 3. Results and Discussion

### 3.1. MALDI-TOF MS Matrix Selection

MALDI matrix selection is important for the desorption and ionization of the test analytes [24]. When MALDI-TOF mass spectrometry was performed, different matrices had distinct desorption and ionization efficiencies, and even matrices themselves may interfere with the detection of test analytes. Furthermore, different types of components might be detected in inverse (positive or negative) ion reflector mode. For example, phenolic acids and flavonoids are easily detected in the negative ion reflector mode, while alkaloids and amino acids are easily detected in the positive mode [1,23,25].

In this work, L-arginine, proline, and phenylalanine were analyzed using DHB, CHCA, and SA in positive ion reflector mode. As shown in Figure 1a, all three matrices could be used for the detection of L-arginine ([M + H]^+^, *m*/*z* 175.1) in the MALDI-TOF MS process, and the intensities of L-arginine ([M + H]^+^, *m*/*z* 175.1) were 9.94 × 10^5^, 1.10 × 10^6^, and 1.57 × 10^5^ using DHB, CHCA, and SA, respectively. The intensities of L-arginine ([M + H]^+^, *m*/*z* 175.1) using DHB and CHCA were significantly higher than those of SA (*p* < 0.05). Besides, the intensities of L-arginine ([M + H]^+^, *m*/*z* 175.1) detected with CHCA as the matrix were significantly higher than that of DHB, however, the complex background peaks interfered with the detection of target analytes (Appendix A). Furthermore, DHB exhibited excellent detection ability in proline ([M + H]^+^, *m*/*z* 116.1; [M + Na]^+^, *m*/*z* 138.1; [M + K]^+^, *m*/*z* 154.1) and phenylalanine ([M + H]^+^, *m*/*z* 166.1; [M + Na]^+^, *m*/*z* 188.1; [M + K]^+^, *m*/*z* 204.1) (Appendix A). Hence, DHB was selected as the most optimal matrix to analyze the spatial distribution of *A. lappa* roots in the positive ion reflector mode.

In previous studies, NEDC, 1,1′-binaphthyl-2,2′-diamine, 1,5-diaminonaphthalene, and 9-AA were used for the detection of flavones and flavone glycosides in the roots of *Scutellaria baicalensis Georgi*, and 9-AA was the optimum matrix due to their stronger ion intensities [25]. In this work, caffeic acid, quercetin, chlorogenic acid, ursolic acid, 1,5-dicaffeoylquinic acid, and 1,3,5-tricaffeoylquinic acid as representative components in *A. lappa* roots were analyzed using DHAP, 9-AA and NEDC in a negative ion reflector mode. All three matrices were able to detect caffeic acid ([M + H]^+^, *m/z* 179.1) and quercetin ([M + H]^+^, *m*/*z* 301.1) (Figure 1b,c). Using DHAP, 9-AA, and NEDC as matrices, the intensities of caffeic acid ([M + H]^+^, *m*/*z* 179.1) were 5.62 × 10^5^, 4.07 × 10^5^, and 1.56 × 10^5^, respectively, and quercetin ([M + H]^+^, *m*/*z* 301.1) were 7.38 × 10^5^, 4.98 × 10^5^, and 5.82 × 10^5^, respectively. DHAP exhibited a significantly higher detection capability in caffeic acid ([M + H]^+^, *m*/*z* 179.1) and quercetin ([M + H]^+^, *m*/*z* 301.1) than 9-AA and NEDC (*p* < 0.05) (Figure 1b,c). Furthermore, DHAP exhibited excellent detection ability in chlorogenic acid ([M − H]^−^, *m*/*z* 353.1), ursolic acid ([M − H]^−^, *m*/*z* 455.2), 1,5-dicaffeoylquinic acid ([M − 2H]^2−^, *m*/*z* 514.8), and 1,3,5-tricaffeoylquinic acid ([M − 3H]^3−^, *m*/*z* 675.6) (Appendix A). Lastly, DHAP has few background interferences, and can be used as a matrix to assist laser desorption/ionization MS imaging to analyze samples (Appendix A). Therefore, DHAP was selected as the best matrix to analyze the spatial distribution of flavonoids and caffeoylquinic acids in the negative ion reflector mode of *A. lappa* roots.

### 3.2. Spatial Distributions of Components in A. lappa Roots

The diameter of *A. lappa* roots is about 1~2 cm with shirred surface, and the moisture content of the root is approximately 70% on wet basis [26,27]. Figure 2 shows fresh *A. lappa* roots, sections of *A. lappa* roots that have been cut, and the tissue structure of the roots. In the MALDI MS imaging process of *A. lappa*, thousands of individual mass spectra data with per shot (200 μm) of *A. lappa* tissue were transformed and reconstructed to present the fusion spectra (Appendix A) and spatial distribution of components on the surface of tissue in the form of images (Figure 3, Figure 4, Figure 5 and Figure 6).

#### 3.2.1. Spatial Distributions of Saccharides in *A. lappa* Roots

Through multiple analyses and comparisons, several similar spatial distribution images in the large molecular weight range (*m*/*z* 500–1700) were found. Thus, it can be speculated that these components might be the same or similar compounds in root tissues. Additionally, there was a 162 Da difference between the adjacent peaks, which meant that these components were glycans, and the molecular weight of 162 Da corresponded to the mass of a hexose residue [28]. In order to verify the presence of saccharide components, 10-fold volumes of water were added to make homogenate, and MALDI MS was used to analyze the components in the homogenate of *A. lappa* with same experimental conditions of MALDI MS imaging. As shown in Figure 3a,b, five peaks and images were obtained in *m*/*z* 527.1, *m*/*z* 689.1, *m*/*z* 851.1, *m*/*z* 1013.1, and *m*/*z* 1175.1 in positive ion mode, which could be speculated to be [DP3 + Na]^+^, [DP4 + Na]^+^, [DP5 + Na]^+^, [DP6 + Na]^+^, and [DP7 + Na]^+^, respectively ([M + Na]^+^). Furthermore, eight similar peaks and images (Figure 3a,c) were obtained in *m*/*z* 543.1, *m*/*z* 705.1, *m*/*z* 867.1, *m*/*z* 1029.1, *m*/*z* 1191.1, *m*/*z* 1353.1, *m*/*z* 1515.1, and *m*/*z* 1677.1, which might correspond to [DP3 + K]^+^, [DP4 + K]^+^, [DP5 + K]^+^, [DP6 + K]^+^, [DP7 + K]^+^, [DP8 + K]^+^, [DP9 + K]^+^, and [DP10 + K]^+^, respectively. However, there were different spatial distributions of saccharides between [M + Na]^+^ and [M + K]^+^, and no saccharides in the cortex ([M + Na]^+^). This might be because of potassium ions were better able to promote the ionization of saccharides than sodium ions, which was related to the distribution and contents of potassium ions and sodium ions in *A. lappa*. Additionally, the peak intensities of these saccharides decreased with the increasing of molecular weight (*m*/*z*) (Figure 3a and Appendix A), which might be because the high-molecular-weight hindered an effective ionization in mass spectrum analysis without any separation or purification process [21]. Therefore, we speculated that the molecular weight of *A. lappa* saccharides could be larger than the present results (DP10) because the section was thin enough (not enough saccharide contents) to limit the ionization of the polysaccharide. Lastly, polysaccharides were the main ingredient of *A. lappa* root, and an inulin-type polysaccharide (2676 Da) was extracted with hot water and identified by high-performance anion-exchange chromatography, methylation analysis, NMR, and other analysis methods in our previous study [6]. Therefore, these peaks might be oligosaccharides or polysaccharides of *A. lappa* roots which were rich in the root center (cambium, pith, secondary xylem, secondary phloem, and cortex) instead of the epidermis (Figure 2 and Figure 3c).

Saccharides are the major bioactive nutrients in food materials or products, including monosaccharides, disaccharides, oligosaccharides, and polysaccharides. They are the most important elicitors to alter physiological and biochemical response in plants, and influence secondary compound productions [29]. Recently, oligosaccharides and polysaccharides obtained from *A. lappa* have received extensive attention, and inulin-type fructan is the major component. Inulin-type fructan exhibited excellent bifidogenic effect, immunomodulatory activity, antioxidant activity, anti-inflammatory activity, and anticancer activity [26,30]. Therefore, it is very important to study saccharides including inulin-type fructan in *A. lappa*. At present, the spatial distribution of saccharides in plants has received increasing attention. It has been reported that hexose and sucrose were found in strawberries using MALDI-TOF MS imaging [18]. Furthermore, there have been some reports supporting that MALDI-TOF MS can be used for analyzing the molecular weight of polysaccharide components, which can provide a high sensitive and rapid analysis strategy for characterizing glycan structures [21]. Furthermore, there also have been some reports regarding the spatial distributions of polysaccharides or oligosaccharides in plant tissues. Feenstra et al. [31] reported that the binary MALDI matrix of organic DHB and inorganic Fe_3_O_4_ could detect large oligosaccharides, and the MS images of hexose polysaccharide (DP 5~10, [M-H_2_O+K]^+^) were obtained from maize seed sections. Sarabia et al. [32] revealed that the oligosaccharides (DP 3~7, [M + Na]^+^ and [M + K]^+^) were mostly in the cortex, while the disaccharides (DP 2, [M + Na]^+^ and [M + K]^+^) were distributed in the whole section of the roots of barley seedlings. This work was the first report regarding the spatial distributions of polysaccharides or oligosaccharides in *A. lappa* roots by MALDI-TOF MS imaging, which could provide ideas for the imaging of carbohydrate polymers and the high-value processing of polysaccharides in *A. lappa* roots.

#### 3.2.2. Spatial Distributions of Caffeoylquinic Acids in *A. lappa* Roots

Caffeoylquinic acids are important acidic metabolites in plants, and their synthesis and amounts are related with several factors including genotype, harvest time, light, and water supplements [33]. Plant roots often secrete these organic acids to regulate the pH value in plants that improve their adaptability to adverse environments [34]. Additionally, caffeoylquinic acids exhibited excellent free radical scavenging ability and protected against oxidative damage, ulcerative colitis, and alleviated hepatic insulin resistance [35,36]. In this work, the caffeoylquinic acids were found successfully using MALDI mass spectrometry imaging analysis, and the image results of representative caffeoylquinic acids were shown in Figure 4. Interestingly, except for caffeic acid, phenethyl eater ([M − H]^−^, 283.3) (Figure 4b) was abundant in the center of *A. lappa* (pith), and other caffeoylquinic acids were rich in the epidermis and cortex including caffeic acid ([M − H]^−^, 179.1) (Figure 4a), D-(−) quinic acid ([M − H]^−^, 191.1) (Figure 4c), and dicaffeoylquinic acid ([M − 2H]^2−^, 514.5) (Figure 4d). Two varieties had similar spatial distribution in same components (caffeic acid and caffeic acid phenethyl). Many more types of caffeoylquinic acids could be detected in the *Baiji* variety, which might be related to composition differences due to different varieties, growing periods, and soil environment.

This work also enriched our previous study that ten caffeoylquinic acid derivatives were obtained by high-speed countercurrent chromatography combined with semi-preparative HPLC from *A. Lappa* roots [8]. Although the MALDI-TOF MS method has been applied to the analysis of caffeoylquinic acid and lignans from the roots and seeds in *A. lappa*, the sample pretreatment using 80% methanol disrupts their spatial distribution and no caffeoylquinic acids were found using the MALDI-TOF MS [1]. Moreover, the detection method of caffeoylquinic acid was improved by using negative ion reflector mode instead of positive ion reflector and DHAP instead of DHB as matrix. This allowed us to obtain information on caffeic acid ([M − H]^−^, 179.1), caffeic acid phenethyl ester ([M − H]^−^, 283.3), D-(−) quinic acid ([M − H]^−^, 191.1) and dicaffeoylquinic acid ([M − 2H]^2−^, 514.5). To identify more caffeoylquinic acids, the suitable matrix for caffeoylquinic acids will be further optimized and developed in future research.

#### 3.2.3. Spatial Distributions of Flavonoids in *A. lappa* Roots

Flavonoids not only play important roles in mediating the response of plants to biological and non-biological environmental stressors as secondary metabolites [37], but also are important ingredients in plants with antioxidant, antibacterial, and hypoglycemic activities [38,39]. It has been reported that the tissues of *A. lappa* contain quercetin, luteolin, quercetin (quercetin and glycoside), flavanol, and other flavonoids [7,40]. However, there was no report related to detect flavonoids in *A. lappa* using MALDI-TOF MS imaging technology.

In this work, quercetin, luteolin, liquirtin, and quercetin-glucoside-rhamnoside were detected in negative ion mode by MALDI MS imaging. Except for quercetin ([M − H]^−^, *m*/*z* 301.2) was distributed in whole tissues (Figure 4e); other flavonoids were abundant in the epidermis in *A. lappa* roots. Luteolin ([M − H]^−^, *m*/*z* 285.2) (Figure 4f) was rich in the cortex near the epidermis and pith of the root tissue. Liquirtin ([M − H]^−^, *m*/*z* 417.4) (Figure 4g) was abundant in secondary phloem. Likewise, quercetin-glucoside-rhamnoside ([M − H]^−^, *m*/*z* 609.5) was distributed in the cortex near the epidermis (Figure 4h). From the distribution of flavonoids in *A. lappa*, the conversion of quercetin to quercetin-glucoside-rhamnoside by UDP-glucose dependent glycosyltransferases might occur in the cortex of *A. lappa* roots [41,42].

Interestingly, the browning degree of the epidermis and cortex was higher than that of the center of the root tissue during our experiments (Figure 2). We observed that the epidermis and cortex were mainly composed of flavonoids and caffeoylquinic acids in the root of *A. lappa*. Therefore, the browning might be due to the reaction of phenolic acids (flavonoids and caffeoylquinic acids) in the root of *A. lappa* without epidermal protection with oxygen in the atmosphere.

#### 3.2.4. Spatial Distributions of Amino Acids in *A. lappa* Roots

Previous studies have shown that the *A. lappa* root contains a variety of amino acids [43]. Using the DHB as a matrix in the positive ion mode, the spatial location of representative amino acid components in the root of *A. lappa* was successfully visualized (Figure 5). Arginine ([M + H]^+^, *m*/*z* 175.1) was abundantly distributed in the secondary phloem of *A. lappa* root (Figure 5a). Similarly, tryptophan ([M + H]^+^, *m*/*z* 205.1) was also found in the secondary phloem with small amounts (Figure 5b). However, the spatial distribution of asparagine ([M + H]^+^, *m*/*z* 133.1) was different from other amino acids, which was mainly distributed in the center of the root tissue (pith) of *A. lappa* (Figure 5c). Moreover, phenylalanine ([M + H]^+^, *m*/*z* 166.1) and proline ([M + H]^+^, *m*/*z* 116.1) were mainly found in the secondary phloem and cambium with a relatively wider range than arginine (Figure 5d,e). Amino acids are the precursors of proteins and some nitrogenous compounds, which play an important role in plant cell signal transduction, root and bud structure regulation, flowering time regulation and stress defense response [44]. In conclusion, the phloem and pith of *A. lappa* might be the main parts for storing energy, coping with biological and abiotic stress, and synthesizing secondary metabolites of *A. lappa* root.

#### 3.2.5. Spatial Distributions of Phospholipids and Choline in *A. lappa* Roots

Phospholipids are the main components of the cell membrane system, involved in cell energy supply, signaling and membrane trafficking in plant immunity [45]. Phospholipids are present in small amounts in most organisms, and the phosphate group of phospholipids is further esterified by serine to form phosphatidylserine [46]. As shown in Figure 5f,g, serine ([M + H]^+^, *m*/*z* 106.1) and phosphatidylserine ([M + H]^+^, *m*/*z* 792.1) had similar distributions, and both existed in the secondary phloem and cambium, indicating that phosphatidylserine should be formed in the phloem and cambium of *A. lappa* root due to the esterification of a phosphate group by serine.

Choline is a precursor participating in plant stress metabolites and is involved in the synthesis of phosphatidylcholine, and is a methyl donor in metabolism [47]. Choline is involved in a variety of metabolic pathways and can be interconverted with phosphatidylcholine, glycerophosphocholine, and phosphorylcholine [48]. In addition, choline is a quaternary amine substance which is extremely basic and suitable for mass spectrometry imaging analysis in positive ion detection mode [49]. The mass spectra imaging results of representative choline were shown in Figure 5h, which showed that choline ([M]^+^, *m*/*z* 104.1) was mainly distributed in the cambium of *A. lappa* root, with a small distribution in xylem, phloem and cortex, and this distribution feature might be related to the biological function of choline.

#### 3.2.6. Spatial Distributions of Unidentified Compounds in *A. lappa* Roots

Four compounds could not be identified because their fragmentation pattern didn’t match with the existing literature and related databases of *A. lappa* root. As shown in Figure 6, the MS imaging results of *m*/*z* 215.0 and *m*/*z* 241.2 ([M]^+^/[M + H]^+^/[M + Na]^+^/[M + K]^+^) in positive ion reflector mode were unidentified, while *m*/*z* 96.9 and *m*/*z* 303.1 ([M − H]^−^) in negative ion reflector mode were unidentified.

Root hairs play important roles in plant autotrophy and fitness, contributing to anchoring root tips into the soil, providing plants with access to water and nutrients from the soil, and interacting with soil microorganisms [50]. Interestingly, a unique component at *m*/*z* 303.1 ([M − H]^−^) was found in the root hair in *A. lappa*. Previous studies have not investigated these unidentified compounds, which may be due to the differences in the variety, origin, planting conditions, climate, and picking time of *A. lappa* tissue, but they confirmed that the MALDI-MS imaging had the potential to explore new components and provide the spatial distribution of components in the field of food science.

### 3.3. Validation Assay

The *A. lappa* of the *Yanagawa-riso* variety was used for the intra-day validation experiment (the experiment was repeated twice, in negative ion reflection mode, with DHAP as the matrix) (Appendix A). The *A. lappa* of the *Baiji* variety was used for the inter-day validation experiment (the experiment was repeated twice, in positive ion reflection mode, with DHB as the matrix) (Appendix A). The results indicated that both slices of the same sample had similar fusion spectra of the endogenous components, demonstrating the good reproducibility of the method. Furthermore, the MALDI-TOF mass spectrometry imaging method in this study does not require complex sample preparation, and the matrix used is stable and can be stored for a long time, which also ensures its reproducibility.

## 4. Conclusions

In the present study, DHB and DHAP were selected as optimal matrices, respectively, in the positive and negative ion reflector mode to analyze components of *A. lappa* roots tissue. Oligosaccharides or polysaccharides, caffeoylquinic acids, flavonoids, amino acids, choline, and phospholipid were imaged by MALDI-TOF MS imaging in the *Baiji* variety and *Yanagawa-riso* variety. The same components were in a similar spatial distribution and different components were found in the root tissue between the *Baiji* variety and the *Yanagawa-riso* variety, which might be related to composition differences due to different varieties, growing periods and soil environment, picking time, etc. Among them, choline, phospholipid, and amino acid were easily detected in the *Yanagawa-riso* variety, while caffeoylquinic acids and flavonoids were easily detected in the *Baiji* variety. Additionally, some compounds observed by MALDI-TOF MS imaging in *A. lappa* were still unidentified and require further study. The increasing interest in using MALDI-TOF MS technologies to map the spatial distribution of components will greatly enrich the development of *A. lappa* roots tissue databases, providing references for developing food processing technologies and the understanding of the plant metabolism of *A. lappa* roots.

## Figures and Tables

**Figure 1 foods-11-03957-f001:**
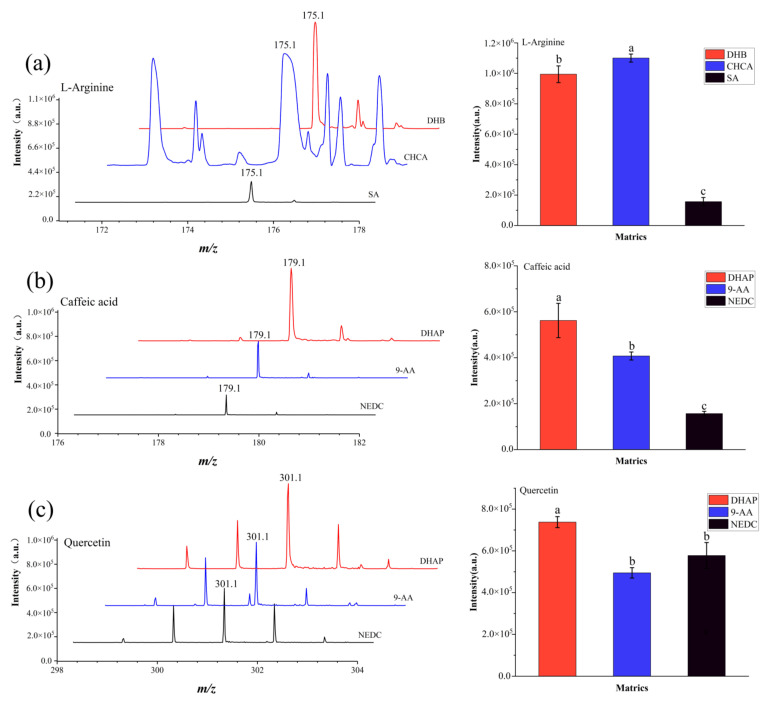
MALDI MS spectra and intensities of representative components using different matrices. MALDI MS spectra and intensities of L-Arginine (**a**) using DHB, CHCA, and SA; MALDI MS spectra and intensities of caffeic acid (**b**), and quercetin (**c**) using DHAP, 9-AA, and NEDC. Different lowercase letters in the histograms indicates significant differences among matrices (*p* < 0.05).

**Figure 2 foods-11-03957-f002:**
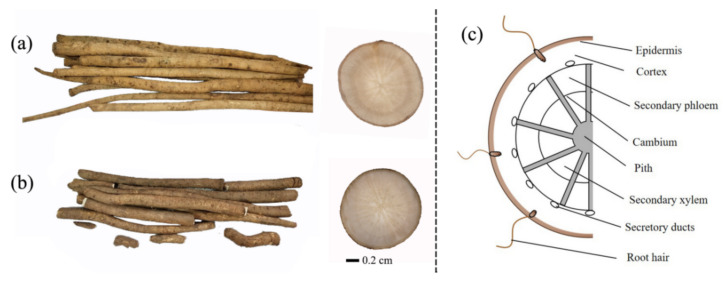
Images and structures of *A. lappa roots*. Fresh *A. lappa* roots and sections cut in *Baiji* variety (**a**) and *Yanagawa-riso* variety (**b**), Author’s collection; and tissue structure (**c**). Adapted from [26], with permission from Elsevier Ltd, 2020.

**Figure 3 foods-11-03957-f003:**
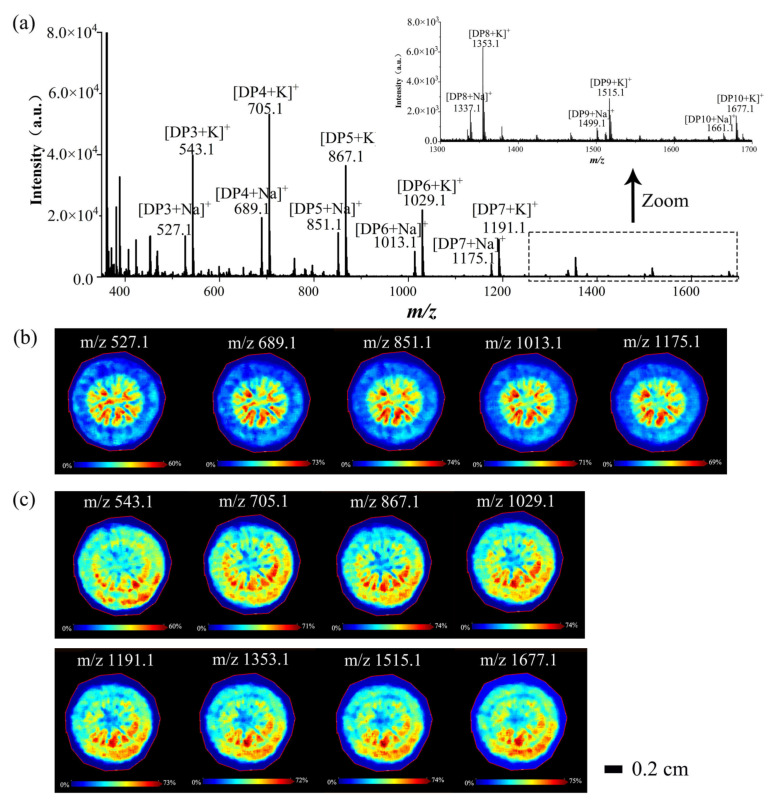
MALDI MS and imaging of saccharides of *A. lappa* roots. MALDI MS (**a**) and images of saccharides in *A. lappa* roots, including [M + Na]^+^ (**b**); [M + K]^+^ (**c**). Data were collected from the *Yanagawa-riso* variety in positive ion reflector mode.

**Figure 4 foods-11-03957-f004:**
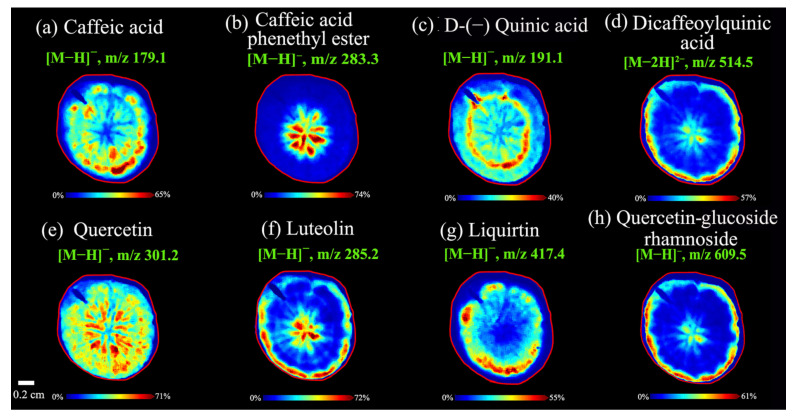
MALDI MS images of caffeoylquinic acids (**a**–**d**) and flavonoids (**e**–**h**) in *A. lappa* roots. Data were collected from the *Baiji* variety in negative ion reflector mode.

**Figure 5 foods-11-03957-f005:**
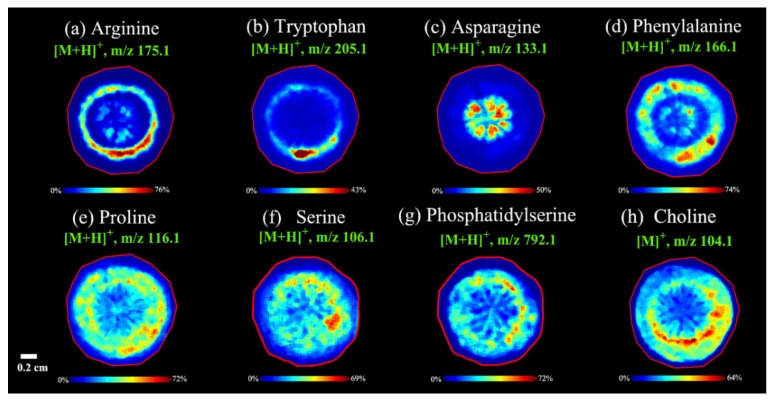
MALDI MS images of amino acids (**a**–**f**), phospholipids (**g**) and choline (**h**) in *A. lappa* roots. Data were collected from *Baiji* variety (**f**,**g**) and *Yanagawa-riso* variety (**a**–**e**,**h**) in negative ion reflector mode.

**Figure 6 foods-11-03957-f006:**
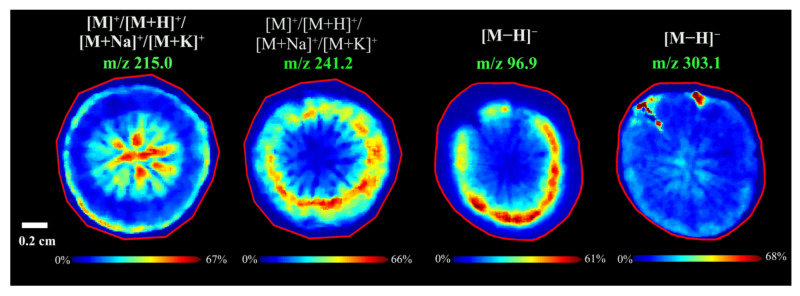
MALDI MS images of unidentified components in *A. lappa* roots.

## Data Availability

Data is contained within the article and Appendix A.

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
