# Peer review of "Visualizing the Spatial Distribution of Arctium lappa L. Root Components by MALDI-TOF Mass Spectrometry Imaging"

_foods, 2022, doi:10.3390/foods11243957_

Round 1
Reviewer 1 Report
In this paper the authors intend to develop novel analytical methods to accurately visualize the spatial distribution of various endogenous components in Arctium lappa roots, and to precisely guide the setting of pre-treatment operations during processing technologies, on the basis of matrix-assisted laser desorption/ionization time-of-flight mass spectrometry (MALDI-TOF MS). Arctium lappa is an important medicinal plant for both nutritional and pharmacological industries. The study revealed that MALDI-TOF MS imaging technology could provide a technical support to understand the spatial distribution of components in A. lappa roots, which would promote the processing technologies for A. lappa roots. The authors are encouraged to improve this manuscript based on minor editorial inputs. For an example, in line 56 the cited reference Liu et al. (2012) should be addressed as Liu at al. [ ] with specific number is square brackets.
Author Response
Dear reviewer:
Thank you for the review and positive comments. Now we have checked the entire manuscript and made corrections accordingly. The detailed corrections have been marked in the manuscript. Please see the revised manuscript.
Sincerely,
Dr. Zhenjia Zheng, corresponding author
Dr. Xuguang Qiao, corresponding author
Reviewer 2 Report
In my opinion, in the presented form the manuscript (foods-1997242) entitled ‘Visualizing the spatial distribution of Arctium lappa L. root components by MALDI-TOF mass spectrometry imaging’ described by Lingyu Li, Zhichang Qiu, Mingdi Jiang, Bin Zhang, Qiang Chen, Chaojie Zhang, Zhenjia Zheng and Xuguang Qiao can be recommended for major revision.
My remarks and recommendations to the Foods are as follows:
The text is comprehensible.
Authors should add data regarding to:
1. Validation is missing (e.g., intra- and interday); and reproducibility.
2. It would be beneficial to supplement the data with data on the effect of the matrix. The authors should estimate the influence of the matrix and complete the data with the 'Index matrix’ or ‘matric effect'.
3. The Authors should clearly indicate what is the scientific novelty of their research. Please indicate clearly what is new with your manuscript for the Foods especially in comparison to earlier of publication(s).
2.2. Matrix selection and MALDI-MS analyses
‘A mixture of DHB/CHCA (1:1, w/w) solution was prepared in 0.1% TFA buffer (ACN/MeOH/water, 70:25:5, v/v/v) at 20 mg/mL as a standard for the calibration of the mass analyzer, and raw spectra were produced with the m/z range of 90–1500.
3. Results and discussion; Figure 3
Please add data / results / spectra for (each) (similar to Figure 3 - in supplementary materials) along with a description of the results:
- matrix / matrices;
- spiked / fortified samples by mixture of standards;
- examples of determined analytes / compounds in the samples.
Were reference materials used?
Reviewer 3 Report
#1997242
The paper Li et al. intitled: “Visualizing the spatial distribution of Arctium lappa L. root components by MALDI-TOF mass spectrometry imaging” presents a study of the detection sensibility of six organic matrices compared to three representative compounds (L-arginine, caffeic acid, and quercetin). Then authors present a the 2D distribution study of radial section of A. lappa root and and Yanagawa-riso variety for the following compounds: amino acids, phospholipids, and choline, caffeoylquinic acids and flavonoids, saccharides. The information of the spatial distribution of the organic matrices would promote and guide the food processing technology of A. lappa roots.
Thanks for this great work with a clear presentation and a well conducted story. The methodological point of the matrix selection and MALDI MS analysis was the hardest to understand for me and also probably of a great interest for the community. Concerning this point, I think the Figure 1 could be improved.
About the gain of the 2D visualization of the distribution of specific compounds, you mentioned it will: ‘provide important value for exploring novel analysis methods in complex food matrices and guiding pre-treatment methods’. Do you think it will also bring new insight more fundamental for the biology of the plant itself about its protein machinery or its energy storage cycle, or about other cycles of the plant? Is it a point to mentioned in the discussion /conclusion ?
I mentioned small corrections or typography in the following.
Suggestions and comments:
#1: Line 212: Figure 3a instead of Fig. 2a
#2: Line 189: I do not understand the different letters above adjacent bars. To modify
#3: Line 314: Cited Fig. 5e

Author Response
Please see the attached file (Response to reviewer 3).
